# Quantitative Thermal Stimulation Using Therapeutic Ultrasound to Improve Cerebral Blood Flow and Reduce Vascular Stiffness

**DOI:** 10.3390/s23208487

**Published:** 2023-10-16

**Authors:** Kyung-Kwon Yi, Chansol Park, Jiwon Yang, Yeong-Bae Lee, Chang-Ki Kang

**Affiliations:** 1Department of Radiological Science, College of Health Science, Gachon University, Incheon 21936, Republic of Korea; 2Department of Health Science, Gachon University Graduate School, Incheon 21936, Republic of Korea; cs11111kr@gachon.ac.kr; 3Department of Neurology, Gil Medical Center, Gachon University College of Medicine, Incheon 21565, Republic of Korea; 4Neuroscience Research Institute, Gachon University, Incheon 21565, Republic of Korea

**Keywords:** therapeutic ultrasound, diagnostic ultrasound, common carotid artery (CCA), cerebral blood flow, pulse wave velocity (PWV)

## Abstract

It is important to improve cerebrovascular health before the occurrence of cerebrovascular disease, as it has various aftereffects and a high recurrence rate, even with appropriate treatment. Various medical recommendations for preventing cerebrovascular diseases have been introduced, including smoking cessation, exercise, and diet. However, the effectiveness of these methods varies greatly from person to person, and their effects cannot be confirmed unless they are practiced over a long period. Therefore, there is a growing need to develop more quantitative methods that are applicable to the public to promote cerebrovascular health. Thus, in this study, we aimed to develop noninvasive and quantitative thermal stimulation techniques using ultrasound to improve cerebrovascular health and prevent cerebrovascular diseases. This study included 27 healthy adults in their 20s (14 males, 13 females). Thermal stimulation using therapeutic ultrasound at a frequency of 3 MHz was applied to the right sternocleidomastoid muscle in the supine posture for 2 min at four intensities (2.4, 5.1, 7.2, and 10.2 W/cm^2^). Diagnostic ultrasound was used to measure the peak systolic velocity (PSV), heart rate (HR), and pulse wave velocity (PWV) in the right common carotid artery (CCA), and the physiological changes were compared between intervention intensities. Compared to pre-intervention (preI), the PSV showed a significant increase during intervention (durI) at intensities of 7.2 W/cm^2^ and 10.2 W/cm^2^ (*p* = 0.010 and *p* = 0.021, respectively). Additionally, PWV showed a significant decrease for post-intervention (postI) at 7.2 W/cm^2^ and 10.2 W/cm^2^ (*p* = 0.036 and *p* = 0.035, respectively). However, the HR showed no significant differences at any of the intensities. The results demonstrate that an intervention at 3 MHz with an intensity of 7.2 W/cm^2^ or more can substantially increase cerebral blood flow and reduce arterial stiffness. Therefore, the use of therapeutic ultrasound of appropriate intensity is expected to improve the cerebral blood flow and reduce vascular stiffness to maintain cerebral blood flow at a certain level, which is closely related to the prevention and treatment of cerebrovascular diseases, thereby improving cerebrovascular health.

## 1. Introduction

Cerebrovascular disease occurs because of cerebral artery occlusion caused by an embolus or thrombus and hemodynamic insufficiency, which interferes with the supply of oxygen and nutrients through the blood. Ischemic stroke, a representative cerebrovascular disease, is highly frequent, accounting for 60–80% of all strokes [1]. Although the mortality rate due to cerebrovascular disease tends to decrease every year as medical technology develops [2], the occurrence of cerebrovascular disease has various aftereffects, including hemiplegia, paresthesia, and visual/linguistic impairment, and lowers the patient’s quality of life despite receiving appropriate treatment [3,4]. Cerebrovascular disease is a fatal disease, with a recurrence rate of 11.3% within a year [5]. As the aftereffects of cerebrovascular diseases impose a burden not only on the family but also on society, it is important to prevent the disease before it develops [6]. Common medical recommendations for the prevention of cerebrovascular disease include smoking cessation, exercise, and diet, all of which increase cerebral blood flow [7]. However, previous studies have shown that these methods must be performed long term, and there are limitations in determining when blood flow changes should be evaluated. Moreover, the effects of these methods are not limited to cerebral blood flow but also affect the overall function of the body. Additionally, it is difficult to apply them uniformly to all individuals, and it is generally difficult to apply it to subjects with various health conditions. Several strategies have been proposed to solve this problem and increase blood circulation, including the electrical stimulation of local areas using transcutaneous electrical nerve stimulation (TENS) [8], electrical muscle stimulation (EMS) [9], and thermal stimulation using therapeutic ultrasound. However, electrical stimulation is not recommended for pregnant women or patients with heart disease [10,11], and incorrect electrical stimulation can cause nerve damage [12]. However, therapeutic ultrasound is used in various fields, including rehabilitation, treatment, and beauty, in recognition of its effectiveness and safety. Therapeutic ultrasound involves adjusting the frequency and intensity to treat injuries such as tumors with heat through high-intensity focused ultrasound (HIFU) [13] and temporomandibular joints with low-intensity pulsed ultrasound (LIPU) [14]. In particular, the use of therapeutic ultrasound to generate thermal stimulation is a safe method that uses microvibrations, which can transfer heat energy from the outside to the body in a noninvasive manner [15], with few side effects [13]. Studies have reported that general hyperthermia treatment reduces arterial stiffness, and the use of therapeutic ultrasound may increase local body temperature and increase blood flow [16,17,18]. However, previous studies have only provided information on tissue blood flow, while there has been insufficient information on interventions with therapeutic ultrasound in the common carotid artery (CCA) and cerebral blood flow to the brain.

Blood flow is a representative indicator of vascular diseases. In the case of the brain, if the supply of cerebral blood decreases below 20 mL of blood per 100 g of brain tissue per minute (mL/100 g/min), speech, sensation, and vision, which play a central role in the brain, can be impaired. If the volume is reduced to less than 10 mL/100 g/min, fatal damage such as necrosis may occur, and nerve loss symptoms, including motor paralysis and cognitive impairment, may appear [19].

Methods for measuring cerebral blood flow include diagnostic medical devices, such as single photon emission computed tomography (SPECT), positron emission tomography (PET), and magnetic resonance imaging (MRI). Both SPECT and PET measure the cerebral blood flow by injecting radiopharmaceuticals into the human body and confirming the degree of drug accumulation, whereas MRI uses brain perfusion and phase-contrast (PC) technology to measure cerebral blood flow. However, SPECT and PET cause radiation exposure, while MRI is sensitive to metals and motion, which limits measurement, and these devices require a long time to obtain data. These limitations can be overcome using diagnostic ultrasound, which has the advantages of being noninvasive and radiation-free, as well as the ability to monitor in real time and infer with blood flow using the measured blood velocity [20].

Pulse wave velocity (PWV) is an indicator of cerebrovascular and cardiovascular diseases and provides information on arterial stiffness and vascular aging. The PWV is the rate at which pulse waves travel to each vascular end due to the contraction of the heart [21]. High PWV is known to increase arterial stiffness and affect stroke [22,23,24]. Methods for measuring PWV include carotid–femoral PWV (cfPWV), which functions by attaching tonometer sensors to the locations below where the carotid artery and femoral artery are located, and brachial-ankle PWV (baPWV), which records pulse waves by attaching a blood pressure cuff to the upper arm and ankle. However, the distance between the two sensors for measuring blood pressure is large, resulting in measurement errors because of inaccurate estimated distance values [25]. In addition, the stiffness of the carotid artery, which is the underlying cerebral vessel, cannot be determined directly. This limitation can be overcome with a one-point measurement method using diagnostic ultrasound, which calculates the PWV using blood pressure and blood vessel diameter after establishing the stiffness index (SI), and there is no error due to the misestimated distance [26,27]. Such a diagnostic ultrasound is characterized by the simultaneous acquisition of blood flow and PWV, which are indicators of cerebrovascular disease.

In this study, we used therapeutic ultrasound to apply a thermal stimulation intervention to the CCA, a blood vessel that supplies blood to the brain, to induce changes in cerebral blood flow, which were then detected using diagnostic ultrasound Doppler-mode scanning. We aimed to develop a stimulation method that can promote cerebrovascular circulation and prevent cerebrovascular diseases by acquiring quantitative information on the changes in cerebral blood flow in response to thermal stimulation. This evidence could also help develop sensors that can measure and detect blood flow and deep body temperature.

## 2. Materials and Methods

### 2.1. Participants

This study was approved by the institutional review board (IRB: 1044396-202207-HR-146-01), and a total of 27 healthy subjects in their 20s participated in this study after obtaining written informed consent. Participants with no history of diseases that could affect cerebral blood flow or the heart and those who were not taking drugs were selected. Prior to participating in the experiment, the participants were restricted from activities that could affect the experiment, such as smoking (6 h), coffee (6 h), and alcohol (12 h), for a period of time (Table 1).

### 2.2. Experiment and Data Acquisition

The participants performed the experiment in the supine position. Diagnostic ultrasound (RS85, Samsung Medison, Seoul, Republic of Korea) and a linear transducer array (LA2-14A, Samsung Medison, Seoul, Republic of Korea) with a frequency bandwidth of 2–14 MHz were used to measure peak systolic velocity (PSV), heart rate (HR), and diameters of the CCA. A smart watch (SM-R850, Samsung Electronics, Suwon, Republic of Korea) was used for blood pressure measurements after calibration for each participant before the experiment. A therapeutic ultrasound (Ultrasound unit US-700, ITO Physiotherapy and Rehabilitation, Tokyo, Japan) with frequencies of 1 and 3 MHz and intensities of 0.5 to 11.0 W/cm^2^ was used for intervention. Therapeutic ultrasonography was performed using a circular drawing around the right sternocleidomastoid muscle over an area of approximately 30 cm^2^ (Figure 1). Considering the intervention depth, a 3 MHz frequency was used [28], and four intensities (2.4, 5.1, 7.2, and 10.2 W/cm^2^) were examined to determine the appropriate stimulus intensity. The order of the intervention intensities was determined randomly, and the participants were unaware of the applied intervention intensity. Before the start of the experiment, the skin was marked after checking the CCA using diagnostic ultrasound so that each measurement could be performed in the same position (Figure 1).

In the pre-intervention (preI) session, diagnostic ultrasound in brightness (B) mode was used to capture images to measure the diameter of the blood vessel for 5 s. Blood pressure was simultaneously measured using a smart watch. After measuring B-mode, images were obtained using pulse wave (PW) and color Doppler (CD) modes to measure PSV and HR for 60 s (‘Acquisition’ in Figure 2). During intervention (durI), acquisition and intervention were performed concurrently for 120 s using diagnostic and therapeutic ultrasounds, respectively. The same measurement as pre-I was performed in the post-intervention (postI) session.

A pilot experiment confirmed that the changes in vital signals in response to the intervention stimulus returned to a normal resting state within 2 min after the intervention; therefore, the time interval between intervention intensities was set to at least 2 min. Considering that the PSV, HR, and blood pressure may be affected by the tension and anxiety of the participants at the start of the experiment, two simulations were conducted in the same manner as in the actual experiment before the start of the experiment to minimize the participants’ errors.

The acquired CCA ultrasound images were stored in DICOM format. DICOM Viewer software (Radiant DICOM Viewer Version. 2021.2.2 (64-bit), Medixant, Poland) was used to record and measure the PSV, HR, and maximum and minimum diameters (MaxD and MinD) of CCA. Both PSV and HR were measured in the preI, durI, and postI sessions and recorded as the average of the measured times. The CCA diameters were measured only in the preI and postI sessions, and both MaxD and MinD were measured based on the inner membrane of blood vessels. MaxD and MinD recorded the largest and smallest diameters in a 5 s image, respectively. Additionally, values that were independently assessed by two radiologists using the same protocol were averaged. Blood pressure was measured only in the preI and postI sessions, and systolic and diastolic blood pressures (SBP and DBP, respectively) were measured to calculate the PWV along with MaxD and MinD. After calculating the SI using Equation (1), the PWV was calculated using Equation (2). Change rate (CR) of the PWV from postI to preI was calculated using Equation (3).
(1)$$Stiffness\;Index\;\left(SI\right)=ln\left(\frac{SBP}{DBP}\right)\times \left(\frac{MinD}{\Delta D}\right),\;\mathrm{where}\;\Delta D\;=MaxD-MinD$$
(2)$$Pulse\;Wave\;Velocity\;\left(PWV\right)=\sqrt{\frac{SI\times DBP}{2\times BD}},\;where\;Blood\;Density\;\left(BD\right)=1.050\;\left[\frac{g}{{\mathrm{cm}}^{3}}\right]$$
(3)$$Change\;Rate\;\left(CR\right)=\left(\frac{postI-preI}{preI}\right)\times 100\left[\%\right]$$

### 2.3. Statistical Analysis

A statistical analysis program (Jamovi version 2.3.18) was used to conduct comparative analysis of the measured values of each session. The reliability of the CCA evaluation results measured by the two radiologists was examined using interclass correlation coefficient (ICC). Repeated measures ANOVA (RM ANOVA) was used to compare the mean differences in PSV and HR between preI, durI, and postI, and variables that showed significant differences were analyzed using post hoc analysis. The mean difference was calculated by subtracting the means, e.g., $\bar{X_{2}}$ − $\bar{X_{1}}$, where the means of factors 2 and 1 are $\bar{X_{2}}$ and $\bar{X_{1}}$, respectively. The standard error (SE) was calculated using the following formula, $\mathrm{SE}=\sqrt{SD_{1}{}^{2}/n_{1}+SD_{2}{}^{2}/n_{2}}$, where the standard deviations and sample sizes of factors 1 and 2 are SD_1_ and SD_2_ and n_1_ and n_2_, respectively. A sphericity test was performed to check for significant differences in the changes in PSV and HR. Greenhouse–Geisser-corrected results were used when sphericity was not satisfied. The level of statistical significance was set at *p* < 0.05 throughout all statistical analyses, after Tukey’s post hoc test. Additionally, the PWV difference between preI and postI was analyzed using a paired *t*-test, and the Bonferroni correction was used for multiple comparison correction.

## 3. Results

The results of the consistency assessment of MaxD and MinD values of the CCA measured by the two radiologists showed high reliability (Cronbach’s α = 0.993).

### 3.1. Peak Systolic Velocity

For PSV at 2.4 W/cm^2^, the difference of durI-preI, postI-preI, and postI-durI was 2.00 ± 1.39 cm/s, 1.89 ± 1.68 cm/s, and −0.12 ± 0.99 cm/s, respectively, and showed no significant difference (*p* = 0.337, *p* = 0.508, and *p* = 0.993, respectively). At 5.1 W/cm^2^, the difference was 2.82 ± 1.30 cm/s, 3.42 ± 2.23 cm/s, and 0.60 ± 1.78 cm/s, respectively, and showed no significant difference (*p* = 0.096, *p* = 0.293, and *p* = 0.940, respectively). Meanwhile, at 7.2 W/cm^2^, the average PSV for preI, durI, and postI was 82.45 ± 20.83 cm/s, 85.99 ± 20.58 cm/s, and 85.99 ± 22.04 cm/s, respectively. The difference of durI-preI, postI-preI, and postI-durI was 3.54 ± 1.12 cm/s, 3.55 ± 1.79 cm/s, and 0.00 ± 1.57 cm/s, respectively, and showed a significant difference on durI-preI (*p* = 0.010, *p* = 0.136, and *p* = 1.000, respectively). At 10.2 W/cm^2^, the average PSV for preI, and durI, and postI was 79.21 ± 19.49 cm/s, 84.16 ± 19.75 cm/s, and 83.27 ± 20.85 cm/s, respectively, and showed a significant difference on durI-preI (*p* = 0.021, *p* = 0.062, and *p* = 0.674, respectively) (Table 2 and Figure 3).

### 3.2. Heart Rate

For HR at 2.4 W/cm^2^, the difference in durI-preI, postI-preI, and postI-durI was −0.27 ± 0.39 cm/s, 0.68 ± 0.55 cm/s, and 0.95 ± 0.56 cm/s, respectively, and showed no significant difference (*p* = 0.774, *p* = 0.442, and *p* = 0.221, respectively). At 5.1 W/cm^2^, the difference was −0.42 ± 0.59 cm/s, 0.40 ± 0.36 cm/s, and 0.82 ± 0.46 cm/s, respectively, and showed no significant difference (*p* = 0.761, *p* = 0.521, and *p* = 0.203, respectively). At 7.2 W/cm^2^, the difference was 0.13 ± 0.46 cm/s, 0.18 ± 0.36 cm/s, and 0.04 ± 0.47 cm/s, respectively, and showed no significant difference (*p* = 0.955, *p* = 0.877, and *p* = 0.995, respectively). At 10.2 W/cm^2^, the difference was 0.58 ± 0.57 cm/s, 0.62 ± 0.58 cm/s, and 0.04 ± 0.51 cm/s, respectively, and showed no significant difference (*p* = 0.568, *p* = 0.543, and *p* = 0.997, respectively) (Table 3). 

### 3.3. Pulse Wave Velocity

Compared to preI at 2.4 W/cm^2^ and 5.1 W/cm^2^, the PWV in postI decreased by 4.17% ± 2.05% and 1.69% ± 1.98%, respectively, but showed no significant difference (*p* = 0.052 and *p* = 0.403, respectively). Meanwhile, at 7.2 W/cm^2^ and 10.2 W/cm^2^, the PWV in postI decreased by 4.79% ± 2.17% and 3.29% ± 1.48%, respectively, and showed a significant difference (*p* = 0.036 and *p* = 0.035, respectively) (Table 4).

## 4. Discussion

Insufficient cerebral blood flow, an indicator of cerebrovascular disease, may lead to insufficient oxygen supply, functional disorders, and an increased incidence of cerebral disease. Another indicator, PWV, is related to arterial stiffness, and a higher PWV may result in an inability of smooth blood flow as arteriosclerosis increases and may also increase the incidence of brain disease [29]. Thus, in this study, changes in blood flow rate (PSV), HR, and PWV according to the therapeutic ultrasound intervention intensity were investigated in the CCA, which is the underlying part of the cerebral vessels.

Comparing the PSV in preI and durI, that in durI was increased at all intensities, it showed no significant difference at a low intensity of 2.4 W/cm^2^ and 5.1 W/cm^2^. However, as the intensity increased, there was a significant difference at an intensity of 7.2 W/cm^2^ and 10.2 W/cm^2^. Additionally, the rate of increase increased with the intensity. The lack of a significant difference may be due to insufficient heat being applied to the tissue at low intensity for 2 min. However, there was a significant difference owing to sufficient heat applied to the tissue at a high intensity for 2 min [30]. In the comparison of preI, the PSV in postI increased at all intensities but showed no significant difference, possibly due to the rapid recovery of the PSV to the normal resting state during postI. Moreover, comparison between durI and postI revealed no significant differences at any intensity. As postI was measured for 1 min immediately after the intervention, it seems that there was an insufficient recovery and/or insufficient time interval to the normal steady state.

Unlike a previous study in which the HR was reduced using therapeutic ultrasound [31], no significant differences were found at any intensity in this study. However, the experimental protocol was different, in that the intervention time was 5 min and the stellate ganglion was directly stimulated. In this study, the intervention time was 2 min, and the sternocleidomastoid muscle was used to avoid superior and middle cervical ganglion stimulation, which is located at the fourth cervical to sixth cervical spine level, to minimize ganglion stimulation [32,33].

Although the PWV decreased at all intensities, no significant difference was observed at low intensities of 2.4 W/cm^2^ and 5.1 W/cm^2^, while a significant difference was observed at high intensities of 7.2 W/cm^2^ and 10.2 W/cm^2^. Previous studies have found that applying external heat to arteries increases arterial compliance [16], which is the inverse of the stiffness index, suggesting that the PWV can decrease, which is consistent with the findings of the current study. This demonstrates that the PWV can be reduced by generating sufficient heat in the tissue using therapeutic ultrasound with an appropriate intensity.

In this study, we divided the intervention strength into four major intensities (2.4, 5.1, 7.2, and 10.2 W/cm^2^), as the first step to find the optimal intensity. We confirmed the changes in the PSV, HR, and PWV when thermal stimulation was applied to the CCA using therapeutic ultrasound. The PSV increased as the intensity increased, and the PWV decreased at all intensities. This is because the higher the intensity, the higher the temperature of the tissue. Accordingly, an increased blood flow velocity [17] and decreased arterial stiffness reflect a decrease in PWV [16]. A decrease in the PWV indicates that the elasticity of the blood vessel increases as the arterial stiffness decreases, so that the blood flow can flexibly respond to the pushing force on the blood vessel wall. The improvements in cerebral blood circulation and reduced arterial stiffness suggest that a certain level of blood and/or nutrients can be smoothly supplied to the brain, contributing to the prevention of cerebrovascular diseases such as hypertension [34]. In the case of the HR, there was no significant difference, and it appeared that thermal stimulation at an appropriate location did not burden the cardiovascular system.

Based on the results of this study, thermal stimulation intervention for CCA can increase the amount of blood flow to the brain, which can be applied to facilitate cerebrovascular circulation and prevent cerebrovascular disease. However, this study has some limitations. First, the participants were small, healthy, and young, so we were unable to provide information about the effects on patients and subjects of different ages. Second, the pre-intervention period was used at the same location as an alternative to sham therapy, but the effects of other locations should be evaluated further. In addition, the measurement of biological signals using the PW and CD modes in durI was limited to measuring the vessel diameter of durI using the B-mode; therefore, it was impossible to confirm the PWV change in durI. Last, it fails to provide information based on various intervention times and frequency changes.

Therefore, further studies are needed to recruit participants from different age groups and patients with diseases to observe significant changes in all groups. The effects of various intervention times should still be compared to determine whether significant differences can be made at low intensities of 2.4 W/cm^2^ and 5.1 W/cm^2^. In addition, it is necessary to check the response when varying the depth of intervention using different frequencies and adjust the postI measurement time and observe changes in biological signals according to the time after the intervention. Finally, the results showed significant differences above 7.2 W/cm^2^. It is necessary to evaluate the subdivided intensities for optimal intensity in a further study. 

## 5. Conclusions

Although there are various methods for improving cerebrovascular health through vascular stimulation, there remains a lack of methods that can quantitatively stimulate the cerebral blood vessels to facilitate cerebral blood circulation. Therefore, in this study, we comparatively analyzed the PSV, HR, and PWV changes in the CCA using therapeutic ultrasound, which is a quantitative method. When interventions were performed using therapeutic ultrasound, PSV showed a significant difference at higher intensities of 7.2 W/cm^2^ and 10.2 W/cm^2^, when comparing preI and durI. PWV showed a significant difference at 7.2 W/cm^2^ and 10.2 W/cm^2^ when comparing preI and postI. However, the HR showed no significant differences at all intensities. This means that interventional intensities above 7.2 W/cm^2^ of existing therapeutic ultrasound can cause significant changes in the PSV and PWV. This also suggests that cerebral blood flow can be easily increased noninvasively without placing a burden on the heartbeat. This suggests that improving the circulation of cerebral blood flow using ultrasonic thermal stimulation intervention may be an important method for improving cerebrovascular health. In addition, the miniaturization of thermal stimulation intervention devices and development of smart wearable sensors that can measure and detect blood flow and deep body temperature will help prevent and reduce the prevalence of cerebrovascular diseases.

## Figures and Tables

**Figure 1 sensors-23-08487-f001:**
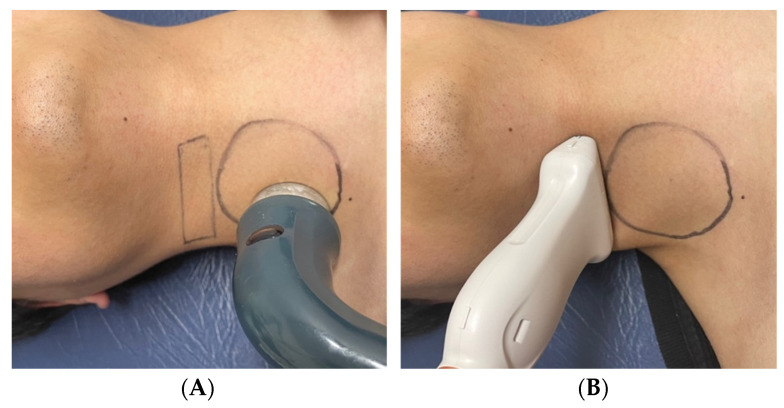
Spots marked for intervention and acquisition: (**A**) Spot for therapeutic ultrasound. (**B**) Spot for diagnostic ultrasound.

**Figure 2 sensors-23-08487-f002:**
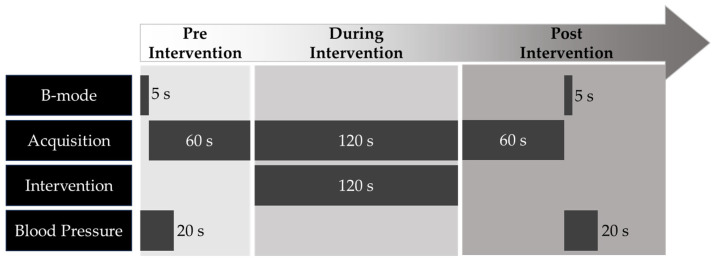
Experimental protocols. The B-mode scan was conducted with a diagnostic ultrasound. The acquisition scans include both pulse wave and color Doppler modes using a diagnostic ultrasound. The intervention was conducted with a therapeutic ultrasound. The measurement of blood pressure was conducted with a smart watch.

**Figure 3 sensors-23-08487-f003:**
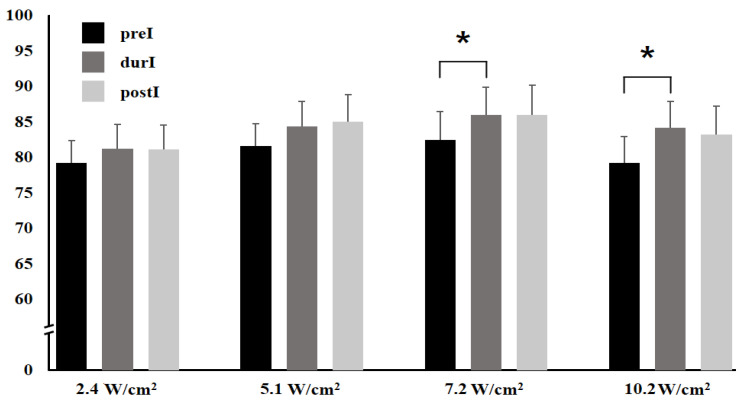
Comparison of peak systolic velocity (PSV) in pre-, during, and post-intervention. * *p* < 0.05.

**Table 1 sensors-23-08487-t001:** Information of participants (mean ± SD).

	Age	Height (cm)	Weight (kg)	BMI (kg/m^2^)
Males (*n* = 14)	21.43 ± 2.10	177.71 ± 6.96	75.93 ± 10.62	24.10 ± 3.49
Females (*n* = 13)	20.62 ± 1.26	163.92 ± 3.86	60.46 ± 5.62	22.49 ± 1.84
Total (*n* = 27)	21.04 ± 1.76	171.07 ± 8.97	68.48 ± 11.53	23.33 ± 2.88

Abbreviations: SD: Standard deviation, BMI: Body mass index.

**Table 2 sensors-23-08487-t002:** Comparison of peak systolic velocity (PSV) using RM ANOVA.

Intensity (W/cm^2^)	F	*p*	Factor 1	Factor 2	Mean Difference ± SE (Factor 2 − Factor 1)	*t*	*p_Tukey_*
2.4	1.32	0.273	preI	durI	2.00 ± 1.39	1.44	0.337
				postI	1.89 ± 1.68	1.12	0.508
			durI	postI	−0.12 ± 0.99	−0.12	0.993
5.1	2.03	0.155	preI	durI	2.82 ± 1.30	2.17	0.096
				postI	3.42 ± 2.23	1.53	0.293
			durI	postI	0.60 ± 1.78	0.34	0.940
7.2	3.64	0.044 *	preI	durI	3.54 ± 1.12	3.17	0.010 *
				postI	3.55 ± 1.79	1.99	0.136
			durI	postI	0.00 ± 1.57	0.00	1.000
10.2	6.01	0.009 *	preI	durI	4.95 ± 1.72	2.87	0.021 *
				postI	4.06 ± 1.70	2.39	0.062
			durI	postI	−0.89 ± 1.04	0.85	0.674

* *p* < 0.05; Abbreviations: SE: Standard error, preI: Pre-intervention, durI: During intervention, postI: Post-intervention.

**Table 3 sensors-23-08487-t003:** Comparison of heart rate (HR) using RM ANOVA.

Intensity (W/cm^2^)	F	*p*	Factor 1	Factor 2	Mean Difference ± SE (Factor 2 − Factor 1)	*t*	*p_Tukey_*
2.4	1.88	0.169	preI	durI	−0.27 ± 0.39	−0.69	0.774
				postI	0.68 ± 0.55	1.24	0.442
			durI	postI	0.95 ± 0.56	1.71	0.221
5.1	1.44	0.248	preI	durI	−0.42 ± 0.59	−0.71	0.761
				postI	0.40 ± 0.36	1.10	0.521
			durI	postI	0.82 ± 0.46	1.76	0.203
7.2	0.09	0.899	preI	durI	0.13 ± 0.46	0.29	0.955
				postI	0.18 ± 0.36	0.49	0.877
			durI	postI	0.04 ± 0.47	0.09	0.995
10.2	0.79	0.459	preI	durI	0.58 ± 0.57	1.03	0.568
				postI	0.62 ± 0.58	1.07	0.543
			durI	postI	0.04 ± 0.51	0.07	0.997

Abbreviations: SE: Standard error, preI: Pre-intervention, durI: During intervention, postI: Post-intervention.

**Table 4 sensors-23-08487-t004:** Comparison of pulse wave velocity (PWV) using a paired *t*-test.

Intensity (W/cm^2^)	Factor 1	Factor 2	Mean Difference ± SE (Factor 2 − Factor 1)	*t*	*p*
2.4	preI	postI	−4.17 ± 2.05	−2.03	0.052
5.1			−1.69 ± 1.98	−0.85	0.403
7.2			−4.79 ± 2.17	−2.21	0.036 *
10.2			−3.29 ± 1.48	−2.23	0.035 *

* *p* < 0.05; Abbreviations: SE: Standard error, preI: Pre-intervention, postI: Post-intervention.

## Data Availability

The data presented in this study are available upon request from the corresponding author.

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
