# Peer review of "Quantitative Thermal Stimulation Using Therapeutic Ultrasound to Improve Cerebral Blood Flow and Reduce Vascular Stiffness"

_sensors, 2023, doi:10.3390/s23208487_

Round 1
Reviewer 1 Report
The study proposes the ultrasound therapy as an innovative intervention in order to increase cerebral blood flow and to prevent cerebrovascular diseases. It is a very interesting subject, apparently a promising future therapeutical method. The study is well designed, the discussions are clearly argumented.
Nevertheless, as limits of the study I would mention the lack of a control group (sham therapy) and the small sample size.
I would also propose some minor corrections:
- Line 47: it should be „the occurrence of cerebrovascular disease has...” instead of „the occurrence of cerebrovascular disease occurs has...”
- Line 61: it should be „transcutaneous electrical nerve stimulation” instead of „translucent electrical nerve stimulation”
The Quality of English Language is quite good, but it could be improved if the paper were reviewed by a professional or native English speaker.
Reviewer 2 Report
I have reviewed the article that the authors have written and I found it to be very well done, written, designed and laid out. I suggest its publication, the studies have important statistical reliability and their procedure is effective.
Round 2
Reviewer 3 Report
The experimental process of this article is complete and the main content is clearly expressed. It is a qualified scientific research article.